# Detection of posttraumatic pneumothorax using electrical impedance tomography—An observer-blinded study in pigs with blunt chest trauma

**Felix Girrbach**[1,2]*, **Tobias Landeck**[2], **Dominic Schneider**[2], **Stefan U. Reske**[3], **Gunther Hempel**[1], **Sören Hammermüller**[1], **Udo Gottschaldt**[4], **Peter Salz**[2], **Katharina Noreikat**[1], **Sebastian N. Stehr**[1], **Hermann Wrigge**[5], **Andreas W. Reske**[1,2,4]

**1** Department of Anesthesiology and Intensive Care Medicine, University of Leipzig, Leipzig, Germany, **2** Innovation Center Computer Assisted Surgery (ICCAS), University of Leipzig, Leipzig, Germany, **3** Heinrich Braun Hospital Zwickau, Department of Radiology, Zwickau, Germany, **4** Heinrich Braun Hospital Zwickau, Department of Anesthesiology and Intensive Care Medicine, Zwickau, Germany, **5** Department of Anesthesiology, Intensive Care and Emergency Medicine, Pain Therapy; Bergmannstrost Hospital Halle, Halle, Germany

* felixfrederic.girrbach@medizin.uni-leipzig.de

**Data Availability Statement:** All relevant data are within the manuscript and its Supporting Information files.

## Abstract

### Introduction

Posttraumatic pneumothorax (PTX) is often overseen in anteroposterior chest X-ray. Chest sonography and Electrical Impedance Tomography (EIT) can both be used at the bedside and may provide complementary information. We evaluated the performance of EIT for diagnosing posttraumatic PTX in a pig model.

### Methods

This study used images from an existing database of images acquired from 17 mechanically ventilated pigs, which had sustained standardized blunt chest trauma and had undergone repeated thoracic CT and EIT. 100 corresponding EIT/CT datasets were randomly chosen from the database and anonymized. Two independent and blinded observers analyzed the EIT data for presence and location of PTX. Analysis of the corresponding CTs by a radiologist served as reference.

### Results

87/100 cases had at least one PTX detected by CT. Fourty-two cases showed a PTX > 20% of the sternovertebral diameter ($PTX_{trans20}$), whereas 52/100 PTX showed a PTX>3 cm in the craniocaudal diameter ($PTX_{cc3}$), with 20 cases showing both a $PTX_{transcc}$ and a $PTX_{cc3}$. We found a very low agreement between both EIT observers considering the classification overall PTX/noPTX ($κ = 0.09$, $p = 0.183$). For $PTX_{trans20}$, sensitivity was 59% for observer 1 and 17% for observer 2, with a specificity of 48% and 50%, respectively. For $PTX_{cc3}$, observer 1 showed a sensitivity of 60% with a specificity of 51% while the sensitivity of

**Funding:** The authors received no specific funding for this work.

**Competing interests:** Hermann Wrigge received research funding, lecture fees, and technical support from Dräger Medical, Lübeck, Germany; funding and lecture fees from InfectoPharm, Heppenheim, Germany; lecture fees from GE Healthcare, Freiburg, Germany, lecture fees from MSD, Konstanz, Germany; and technical support from Swisstom Corp., Landquart, Switzerland. The other authors declare that they have no competing interests.

**Abbreviations:** ARDS, Acute Respiratory Distress Syndrome; COPD, Chronic obstructive pulmonary disease; CT, Computed Tomography; EIT, Electrical Impedance Tomography; HU, Hounsfield Units; PTX, Pneumothorax; ROI, Region of Interest.

observer 2 was 17%, with a specificity of 89%. By programming a semi-automated detection algorithm, we significantly improved the detection rate of $PTX_{cc3}$, with a sensitivity of 73% and a specificity of 70%. However, detection of $PTX_{transcc}$ was not improved.

## Conclusion

In our analysis, visual interpretation of EIT without specific image processing or comparison with baseline data did not allow clinically useful diagnosis of posttraumatic PTX. Multimodal imaging approaches, technical improvements and image postprocessing algorithms might improve the performance of EIT for diagnosing PTX in the future.

## Introduction

Pneumothorax is a common complication after blunt chest trauma [1]. Undiagnosed, occult pneumothoraces may convert to tension pneumothorax—particularly after initiating mechanical ventilation—and require emergency thoracostomy [2,3].Tension pneumothorax is still one of the main causes of preventable deaths following multiple trauma [4].

In the in-hospital setting, chest X-ray (CXR) is often the initial imaging technique used for confirming or excluding pneumothorax after blunt chest trauma. However, recent publications revealed that the prevalence of occult pneumothoraces—not visible in the anteriorposterior CXR–is high [5]. This applies especially to chest trauma patients, in whom the rate of occult pneumothoraces can reach 55 to 69% [2,5,6].

Many authors thus advocate computed tomography (CT) as the gold standard for detecting pneumothorax in patients with severe chest trauma [7,8] because of its higher sensitivity compared to chest X-ray [5,9]. However, CT involves significant radiation exposure. CT is not availabe at bedside and transportation to the radiology department imposes additional risks. In most prehospital settings, CT is not available at all.

Because of such limitations of CT, there has been an ongoing evaluation of alternative diagnostic modalities for detecting pneumothorax, which can be used at the point of care, repeatedly or even continuously, and without radiation exposure. One such technique is thoracic ultrasound, whose sensitivity and specifity are outperforming CXR [10]. Despite many advantages, sonography provides only limited information about the dimension of the pneumothorax, is highly observer dependent [10] and can be influenced by comorbidities like chronic obstructive pulmonary disease (COPD) and acute respiratory distress syndrome (ARDS) [11].

Another thoracic imaging technique that shares advantages such as non-invasiveness and mobility is Electrical Impedance Tomography (EIT). For diagnosing PTX, EIT and sonography may actually provide complementary information.

For EIT, an electrode belt containing 16 or 32 electrodes is placed around the chest cranial to the diaphragm at the height of the 4th to 6th intercostal space. A defined alternating current (typically 5mA at a frequency of 50 kHz) is applied to a first pair of electrodes and the resulting surface potentials are registered in the remaining electrode pairs. The location of the current injection and voltage measurements is rotated continuously around the chest. Thus, a complete rotation of the injecting electrode pair results in 16 voltage profiles and a total of 208 voltage measurements when using a 16-electrode belt. Bioimpedance is calculated using Ohm's law. After filtering and algorithm-based reconstruction in a processing unit, a two-dimensional, real-time cross-sectional image of pulmonary ventilation is displayed on a monitor.

Changes in relative impedance during a respiratory cycle are displayed using a white to dark blue color scheme.

Without involving radiation, EIT is able to display regional ventilation changes within the lung in real-time [12]. Several authors have recently reported the use of EIT for pneumothorax detection and were able to show encouraging results [13–15]. However, a major limitation of these previous studies is the dependence on comparison with a baseline EIT measurement obtained under healthy lung conditions, before development of PTX. Availability of such a baseline EIT data appears realistic only in very few clinical situations. Whether currently available EIT technology could aid in the more common situation that diagnosis or exclusion of a pneumothorax is sought in a patient for whom no reference information is available, remains unclear.

We therefore designed this study to test the hypothesis that pneumothorax can be detected by visual analysis of EIT images without additional information from pre-existing reference images or application of specific image processing.

## Material and methods

In compliance with the 3R's for reduction of animals in research (https://www.nc3rs.org.uk/the-3rs), we used suitable data from an existing institutional database in this experimental, observer-blinded diagnostic study. The original experiment was approved by the governmental animal ethics committee (Landesdirection Leipzig, reference number TVV38/11). Pigs (German Landrace) were received from a conventional pig-breeding farm and were kept at the Large Animal Clinic for Surgery, Faculty of Veterinary Medicine, University of Leipzig in a species-specific environment.

From this database, we randomly obtained 100 EIT files and the corresponding CT data from 17 different pigs. The files were anonymized and observers were blinded to all other information. Two observers (specialists in anesthesiology and intensive care) analyzed the randomly obtained 100 EIT files, while a third observer (specialist in radiology) analyzed the corresponding CT images.

### Animal preparation and induction of chest trauma

Animals were sedated with intramuscular ketamine and midazolam. After induction of anesthesia, the airway was secured by surgical tracheostomy. Central venous, arterial, pulmonary arterial and urinary bladder catheters were placed under sterile surgical conditions. Throughout the experiment, the animals were continuously anesthetized with sufentanil, ketamine and midazolam and mechanically ventilated using tidal volumes of 6 ml/kg actual bodyweight. Muscle relaxation was maintained by continuous infusion of pancuronium throughout the experiment. Usual intensive care support was provided as necessary (i.e. antibiotics, IV fluid support, vasopressors).

After completion of instrumentation and induction of experimental chest trauma (see below), all animals were placed in supine position on the CT table for the entire duration of the experiment.

For induction of blunt chest trauma, the animals were positioned on their left side, and the EIT belt was removed after carefully marking its position. A 10 kg steel weight was dropped through a tube on the right hemithorax from a height of 1.80 m. Immediately following induction of the trauma, CT was performed and chest drains were inserted into pneumothoraces that exceeded 1/3 of the thoracic diameter or showed signs of tension pneumothorax. Because continuous suction was not applied to the chest tubes, pneumothoraces persisted despite drainage and application of PEEP in most cases. Pneumothorax persisted particularly often in

the accessory lobe (a specific anatomical structure in pigs) because this region is covered by additional pleural layers and thus difficult to reach with chest drains.

For the original study, the animals were randomly assigned to three different ventilation strategies after induction of the chest trauma: ARDS-Network low-PEEP protocol [ARDSNet [16]], Open-Lung-Concept [OLC, [17]], or an EIT-based ventilation strategy [18,19]. If necessary, ventilator settings were adapted every 4 hours after measuring arterial blood gases and according to the designated strategy. At the end of the experiment, animals were euthanized intravenously by injection of 2 grams thiopental and 50 ml of 7.45% potassium chloride (1 molar) according to our local protocol.

For each individual, several standardized measurements (baseline, post-trauma, every 4 hours for 24 hours) were available in the database, each including a three minute EIT recording and a corresponding full chest CT scan. Due to the high frequency inverse ratio ventilation of OLC, animals of the OLC-group were excluded and only images obtained from animals from the ARDSNet and the EIT arm were included in this study.

## Acquisition and analysis of CT images

CT scans were performed in expiratory hold and images were reconstructed with 6 mm slice thickness and the standard (B) reconstruction kernel (Mx8000 IDT 16, Philips Healthcare, Hamburg, Germany). CT scans selected from the database were analyzed by a radiologist with extensive experience in trauma imaging using the Medical Imaging Interaction Toolkit (MITK v2016.11–3, German Cancer Research Center, Heidelberg, Germany). The scans were displayed at appropriate window settings (-350/1350 Hounsfield units, HU). The CT slices showing the greatest number of electrodes of the EIT-belt and an additional two slices (3 cm cranially and caudally, respectively) were analyzed. The number of pneumothoraces and their distribution over the four quadrants of the thoracic cross-sectional areal (i.e. ventral right, ventral left, dorsal right, dorsal left quadrant) were recorded on CT analysis sheets. Likewise, pleural fluid collections and atelectasis were sketched on the CT analysis sheet. Additionally, the size of every visible pneumothorax was calculated using the respective MITK functionality and documented on the CT analysis sheet.

In a second step of CT image analysis, segmentation of the three selected CT slices mentioned above was performed manually using the MITK segmentation tool, using different colors for PTX, the borders of the lung as well as the thoracic contour. The three slices were then converted to one single summary image (slab) by manually overlapping the three slices as depicted in **Fig 1A** using the 3D visualization tool of MITK. This step was performed to include three-dimensional information concerning the cranio-caudal expansion of the pneumothorax. Furthermore, the CT-slabs were created to imitate the rather elliptic, three-dimensional information given in the EIT image for better comparability.

## Acquisition and analysis of EIT data

The EIT system (PulmoVista 500, Dräger Medical, Lübeck, Germany) with a belt that carries 16 electrodes was attached just caudal to the animal's axillae. The belt remained in the same position for the duration of the experiment, with exception of the short period for the induction of the chest trauma. Furthermore, the electrode positions were marked on the skin to allow correct repositioning of the belt during the 24-hour experiments if necessary. EIT files were recorded at 50 Hz frame rate and stored on an external hard disk for further off-line analysis.

After retrieval from the database, EIT files were evaluated for potential pneumothoraces by two observers blinded to the CT results and any additional information beyond the EIT image

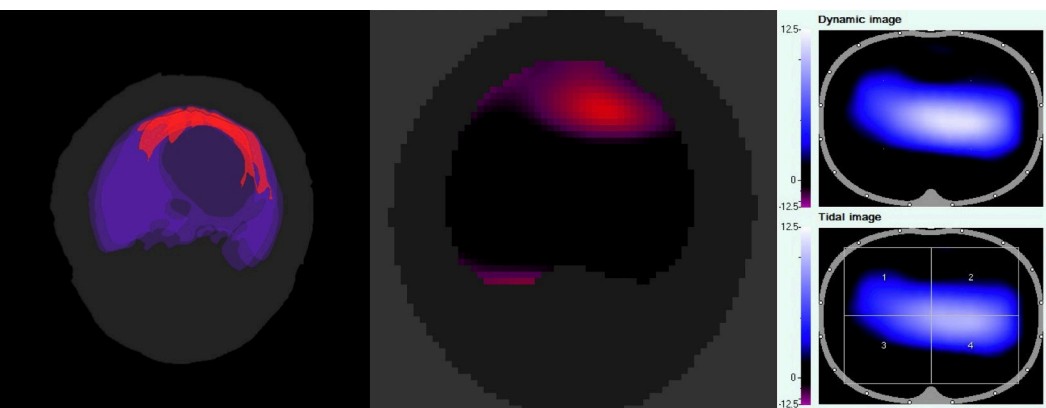

**Fig 1. Comparison of CT, EIT detection algorithm and EIT monitor for detection of a PTX in one exemplary dataset of the database.** A. CT visualization of a clinical relevant PTX after segmentation of the PTX and conversion of the three slices to a single, color coded image (CT slab). Light grey: thoracic contour, blue: lung contour, red: PTX. B. The same PTX visualized by the newly developed, semi-automatized detection algorithm (see methods section). The PTX is also marked in red color. C. Corresponding EIT file. Note that ad-hoc detection of the PTX is not possible in the unprocessed EIT file.

data. Both EIT-observers were specialists in anesthesiology and intensive care medicine and experienced in the use and interpretation of EIT. Observers could watch the dynamic EIT image, the tidal image (i.e. the differential image between end of inspiration and expiration) and the impedance waveform tracing of the entire thorax and/or thoracic quadrants. The observers were allowed to watch the EIT files repeatedly if necessary. The graphical user interface of the EIT Data Analysis Tool (version 6.1, Dräger, Lübeck, Germany) was used for visualization. No specific changes were made to this software. The EIT image reconstruction baseline was automatically determined by the software. Similar to the CT analysis, the two EIT observers used an EIT analysis sheet for documentation of suspected pneumothoraces and their location according to the quadrants mentioned above. Because the aim of this study was to assess the capability of EIT for *ad-hoc* visual identification of pneumothorax, we refrained from using specific detection algorithms or image processing at this step. Beyond evaluating the EIT images for defects in ventilation, investigators considered several characteristics of EIT images that have already been described to be associated with a possible (tension) pneumothorax: (1) presence of high-impedance spots with either an asynchronous behavior in comparison to the tidal impedance changes of the remaining lung or which were not ventilated at all; (2) asymmetric distribution of impedance changes in the dynamic or tidal image [15]; (3) a progressive decrease in regional ventilation (i.e. impedance amplitude) [13]; (4) presence of spike-like patterns in the impedance waveform tracing of the entire thorax and/or thoracic quadrants [20].

## Comparison of visual EIT interpretation and CT results

Based on CT, PTX were arbitrarily defined as clinically relevant if they had a craniocaudal size of $> 3$ cm ($PTX_{cc3}$), if their maximum transversal diameter exceeded 20% of the animal's sternovertebral distance ($PTX_{trans20}$), or both ($PTX_{transcc}$). The sternovertebral distance was measured at the CT slice depicting the most electrodes of the EIT belt. Smaller PTX were excluded from the analysis because they were considered to be beneath the spatial resolution of current EIT hardware. Because PTX located in the accessory lobe were difficult to classify using the 4 ROIs, and for more precise information of the location of the PTX, we determined the location of the largest PTX visible in the summary image with the aid of a dedicated classification sheet

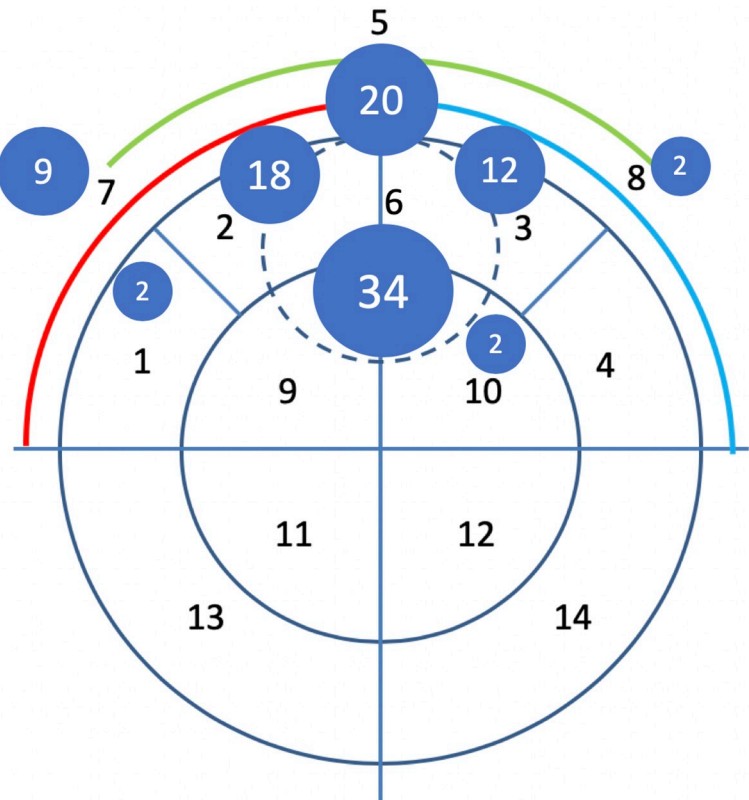

**Fig 2. Newly developed PTX classification for the classification and localization of PTX in pigs.** The PTX classification was developed to cope with the special pulmonary anatomy in pigs with an accessory median lobe (segment 6) where most of the PTX were observed in the study. The black numbers indicate the number of the defined segment, the white numbers in the blue circles indicate the quantity of observed PTX in the specific segment. Note that most PTX were observed in the anterior segments of the lung. Segment 5, 7 and 8 are combinations of 2 segments.

that respected the slightly different pulmonary anatomy of pigs in comparison to humans (**see Fig 2**). The drawings on the EIT scoring sheet were analyzed in the same way and the results were transferred to an Excel spreadsheet (Microsoft Corp. Redmond, CA, USA) for further statistical analysis.

### Development of a semi-automatized diagnostic algorithm

An automatized diagnostic algorithm was developed in MatLab (Mathworks, Natick, MA, USA) using pooled EIT data of 20 healthy, mechanically ventilated pigs (EIT data obtained at the baseline measuring point from the initial experimental study). The program was designed to compare the actual EIT file with the pooled information of healthy controls and the region of a suspected PTX was visualized by the program in the tidal image (see **Fig 1B**). The region of the suspected PTX was also classified using the PTX classification sheet depicted in **Fig 2**.

### Statistical analysis

Data were transferred to Excel spreadsheets (Microsoft Corp. Redmond, CA, USA) and analyzed using SPSS 17 (IBM, Ehingen, Germany). Data are given as mean and standard deviation (SD) unless otherwise stated. Agreement between the two EIT-observers as well as between each of the EIT-observers and the CT results was determined by calculating the Cohen's

Kappa-coefficient for the following comparisons: First, EIT and CT results were compared using the binary information (pneumothorax-positive / pneumothorax-negative). Subsequently, interobserver and observer/CT agreement, as well as the agreement between the automatized EIT-algorithm and CT imaging, were calculated based on the semiquantitative classification described above. Positive and negative predictive values, sensitivity and specificity were calculated for both EIT-observers and the automatic detection algorithm, with the CT data serving as reference. The resulting κ-values were interpreted as follows: κ < 0.2: no agreement, $0.2 \leq \kappa < 0.4$ minimal agreement, $0.4 \leq \kappa < 0.6$ weak agreement, $0.6 \leq \kappa < 0.8$ strong agreement, $0.8 \leq \kappa < 0.9$ very strong agreement, $0.9 \leq \kappa \leq 1.0$ almost perfect agreement. P-values <0.05 were considered to be statistically significant.

## Results

The corresponding CT and EIT image pairs originated from 17 different animals of the original experiment at different time points, with a minimum of 1 EIT/CT dataset and a maximum of 11 datasets per animal. Including all PTX visible in at least one of the 3 CT slices, 87/100 cases were PTX positive, with a median of 2 PTX in each of the 100 image sets (range 0–7 PTX/set).

Fourty-two of 100 cases showed PTX > 20% of the sternovertebral diameter ($PTX_{trans20}$), whereas 53 of 100 cases showed a PTX > 3 cm in the craniocaudal diameter ($PTX_{cc3}$). There were 4 cases with a PTX that exceeded 3 cm in their craniocaudal diameter as well as > 20% of the sternovertebral diameter ($PTX_{transcc}$). These PTX were located in the segments 3, 5, and 7 (2 cases). In 20 cases, both had a $PTX_{cc3}$ and a $PTX_{trans20}$ that were considered independent from each other. All clinically relevant PTX were located in the nondependent lung quadrants (segments 1–10, respectively, see also **Fig 1**).

### EIT analysis compared to CT imaging

For $PTX_{trans20}$, sensitivity was 59% and 17% for observer 1 and observer 2, with a specificity of 48% and 50% respectively. For $PTX_{cc3}$, observer 1 showed a sensitivity of 60% with a specificity of 51% while the sensitivity of observer 2 was 17%, with a specificity of 89%.

There was a very low agreement between EIT analysis and CT imaging for both observers considering the classification PTX/noPTX:

Fourteen $PTX_{trans20}$ (33%) were missed by both EIT observers, with 13 of these located in segment 6. One example is shown in **Fig 3**. Likewise, 19 of 53 $PTX_{cc3}$ (36%) were missed by both observers, with 18 of these (95%) being located in the anterior segments 2,3 and 5.

### EIT Inter-observer reliability

Although both observers located most of the PTX in the anterior segments (2, 3, 5 and 6) of the EIT images (95% for observer 1; 100% for observer 2) there was very low agreement between the two observers in the binary classification of PTX/noPTX when all the datasets are considered (κ = 0.09, p = 0.18). EIT observer 1 rated 55 of the 100 EIT-image datasets PTX positive, whereas EIT observer 2 rated 14/100 cases as PTX positive.

We identified a total of 10 of the 100 EIT-image datasets where both observers agreed in the rating as "PTX positive". All of these PTX were suspected by the observers to be in the same or in neighboring, anterior pulmonary segments (segments 2, 3, 5, 6). Further analysis of these 10 EIT-image datasets revealed a PTX in the corresponding CT image, with 8 of the cases showing a clinically relevant, anteriorly located PTX. In nine of these cases, the EIT files showed an end-expiratory signal or a region with asynchronous ventilation in the region of the suspected PTX. One of the 10 EIT-image datasets was classified as PTX positive by both observers due to

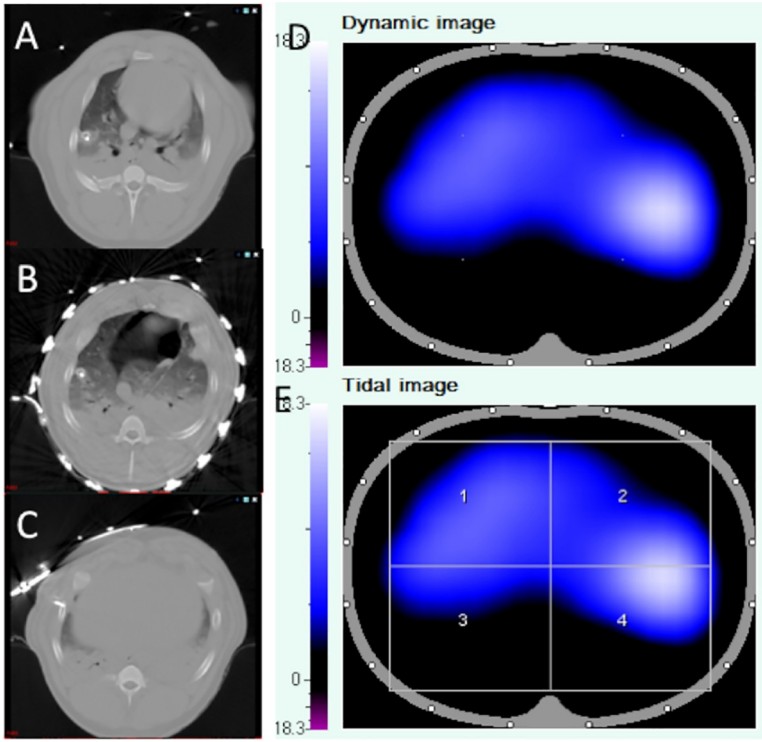

**Fig 3. Example of a PTX$_{transcc}$ not detected by both observers due to a partial volume effect.** The CT images at the level of the belt (B), and three centimeters cranial (A) and caudal (C) to the EIT belt show a subcardial PTX. However, the PTX is not visible in the EIT, neither in the corresponding dynamic nor the tidal EIT image. This may be explained by a "partial volume effect", which means that the higher conductibility of the surrounding soft tissue (diaphragm, heart) partly counterbalances the low conductibility of the PTX.

a region with asynchronous ventilation mimicking a PTX. However, in this case, CT only revealed a very small PTX remote from the site where it was suspected by the EIT-observers.

## Automatized detection algorithm vs. CT imaging

For PTX$_{trans20}$, sensitivity of the detection algorithm was 45%, with a specificity of 42%. PPV was 37% and NPV was 51%. The detection algorithm also showed a poor agreement compared to CT analysis ($\kappa$ = -0.1, p = 0.21).

For PTX$_{cc3}$, the detection algorithm had a sensitivity of 73% and a specificity of 70%, resulting in a PPV of 73% and NPV of 68%. There was a fair agreement to CT imaging for PTX$_{cc3}$ ($\kappa$ = 0.4, p<0.01).

All PTX$_{transcc}$ (4/4, 100%) were correctly identified by our detection algorithm. A summary of the performance of the detection algorithm compared to observer 1 and 2 is given in **Tables 1 and 2**.

**Table 1. Number of correctly classified PTX, false positively classified PTX and false negatively classified PTX according to the observer/detection algorithm.**

|  | Correctly classified (%) | Classified false positive (%) | False negative (%) |
|---|---|---|---|
| **Overall relevant PTX (n = 71)** |  |  |  |
| Observer 1 | 60 | 12 | 28 |
| Observer 2 | 37 | 3 | 60 |
| Detection algorithm | 57 | 12 | 31 |

**Table 2. Number of PTX that were missed by the EIT observers and the detection algorithm according to their size.** PTXcc3: PTX > 3 cm in craniocaudal diameter, PTXtrans20: PTX > 20% of the sternovertebral diameter, PTXtranscc: PTX > 3 cm in craniocaudal diameter and > 20% of the sternovertebral diameter.

| Category | n | PTX missed by both observers (n) | % missed by both observers | % missed by detection algorithm |
|---|---|---|---|---|
| Overall relevant PTX | 71/100 | 24/71 | 33% | 31/71 (34%) |
| PTXcc3 | 53/100 | 22/53 | 42% | 15/53 (28%) |
| PTXtrans20 | 42/100 | 14/42 | 33% | 23/42 (55%) |
| PTXtranscc | 4/100 | 1/4 | 25% | 0/4 (0%) |

## Spike potentials

EIT-observer 1 detected 51 spike-like formations in the cyclic ROI information (**Fig 4**) whereas EIT-observer 2 only characterized 13 of the 100 cases as spike potentials, also showing a poor inter-rater reliability ($\kappa$ = 0.172, p = 0.009). For both observers, there was no agreement between the detection of spike potentials and presence of a PTX (observer 1: $\kappa 1$ = -0.02 [p = 0.87], $\kappa 2$ = 0.08 [p = 0.43]; observer 2: $\kappa 1$ = 0.03 [p = 0.75], $\kappa 2$ = 0.08 [p = 0.21], $\kappa 1$ for $PTX_{trans20}$ and $\kappa 2$ for $PTX_{cc3}$, respectively).

However, in 11 cases, both EIT observers detected spike-like potentials in the EIT plethysmograph. All 11 cases revealed a PTX in the CT analysis which was at least partially located between the heart and the parietal pleura, allowing contact between the pericardium and the inner thoracic wall. Further analysis of the spike potentials in these cases revealed that the frequency of spike potentials precisely matched the animal's heart rate.

## Discussion

Experimental evaluation of EIT has resulted in crucial insights into lung pathophysiology and hence, several experts have advocated broader clinical application of EIT. In this context, the use of EIT for pneumothorax detection has been among the most promising clinical applications. To our knowledge, our study is the first observer-blinded study investigating the potential of EIT to detect a pneumothorax *ad-hoc*, i.e. without a pre-existing reference image of a healthy lung.

Currently, sonography of the chest is a valuable bedside tool for the diagnosis of a PTX. However, in the case of suspected PTX, sonography requires an experienced operator and does not provide information on the size of the PTX because total reflection of ultrasound waves occurs at tissue-air interfaces. Additional radiologic imaging using ionizing radiation is therefore often performed to evaluate the size of a known or sonographically suspected PTX.

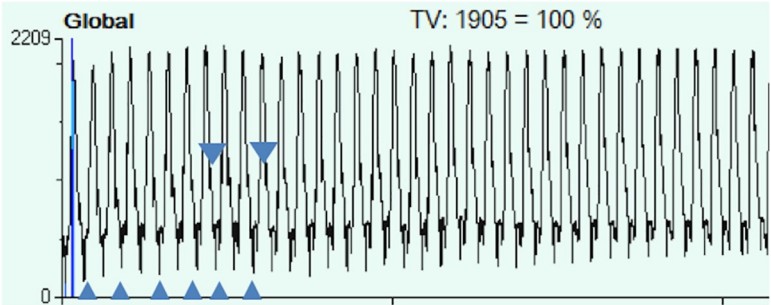

**Fig 4. Spike potentials in the cyclic ROI information.** Examples of spike potentials are marked by triangles. The observed spike potentials show sudden drops in thoracic impedance at a frequency corresponding to the animal's heart rate and my therefore be explained by cyclic contact of the heart and the chest wall during systole, which improves thoracic electrical conductivity.

Because EIT is able to display ventilation defects, the combination of demonstrating the absence of pleural sliding by ultrasound and showing a ventilation defect in the EIT image may provide enough indication that decompression of a PTX is necessary–without the additional need of ionizing radiation. Nevertheless, literature is sparse concerning the detection of pneumothoraces using EIT [13–15,21].

In our study, the EIT observers grossly differed in their perception whether a PTX was visible on the EIT file. Although both EIT-investigators were experienced with EIT-monitoring and interpretation in humans, the different pulmonary anatomy of pigs (e.g. lobus accessories located ventromedially) might have led to misinterpretations of ventilation defects. The agreement between both EIT observers and CT did not even reach levels expected by chance.

One explanation is that EIT is only able to deliver *functional* rather than morphological images of the lung. Ventilation defects may result from different intrathoracic structures or pathologies. Without additional imaging data, it seems difficult to differ between diverse pathologies such as atelectasis, hemothorax, or pneumothorax. This explanation is partly alleviated by the fact that all PTX in our study considered relevant in size were in the nondependent part of the lung and therefore atelectasis or hemothorax would not be a relevant differential diagnosis in these cases.

Our analysis also showed that detection of a PTX without a preexisting reference image might additionally be limited by the currently low spatial resolution of EIT, which seems to be too low for the detection of smaller pneumothoraces. Smaller PTX—particularly concerning the craniocaudal diameter—are subject to what we called a "partial volume effect" (**Fig 3**). This means that summation of areas with low impedance may partially blur the contours of areas with high impedance and therefore make identification of a PTX more difficult for the observer.

The ability of EIT to detect a PTX may also depend on the behavior of the PTX. Asynchronous ventilation might only be seen as long as there is movement of gas in the PTX. High PEEP levels might prevent relevant movement of gas entrapped in the PTX, which may thus not be visible. However, this suggestion—as well as the behavior of PTX in the setting of different PEEP levels—warrants further investigation.

By programming a simple detection algorithm which uses pooled data of healthy pigs for comparison and to identify areas of abnormal or missing ventilation, we were able to significantly improve the detection rate of PTX compared to a solely visual analysis. Interestingly, only the detection rate of PTX > 3 cm in craniocaudal diameter could be improved whereas the detection rate of PTX with large transverse diameter, but small craniocaudal diameter, could not be improved. Again, this finding supports the assumption that PTX diagnosis by EIT is limited by a partial volume effect due to the elliptic dissemination of the injected current in the chest. However, the clinical relevance of this limitation may be questioned as clinically relevant PTX in humans usually have a craniocaudal diameter exceeding 3 cm, especially in the absence of pleural adhesions. Further improvements of image reconstruction and spatial resolution of the EIT may also improve this issue in the future.

In a completely different clinical scenario, Costa et al. [13] as well as Bhatia and colleagues [14] could locate a pneumothorax with a minimal volume of 10–20 ml with 100% sensitivity in animals using specific detection algorithms, which compared regional ventilation changes to the respective individual healthy baseline situation. While these detection approaches have obvious limitations for pneumothorax detection when baseline data is not available, but they can still be very helpful in high risk situations, where pneumothorax development and/or progression are likely and monitored by continuous EIT. Among such situations are high airway pressures in patients with COPD or patients in which problems (i.e. air aspiration) occurred during an attempt to insert a central venous catheter [21,22].

Cambiaghi et al [20] stated in their recently published case report that the analysis of "spiky patterns" in the course of electrical thoracic impedance could be an early sign of a developing pneumothorax. We were not able to confirm these results, maybe because of the lack of an objective definition of spike potentials. Moreover, we were able to show that cardiac activity may also lead to spike potentials in the cyclic ROI information curve. These findings are in line with the theory of Cambiaghi et al, in which they state that spike potentials occur when there are sudden changes of thoracic impedance. While Cambiaghi *et al*. observed a sudden rise in thoracic impedance due to a pulmonary air leak with air entry into the PTX, we observed a sudden, cyclic drop in thoracic impedance with a frequency of the animal's heart rate in some of the animals. In the 11 cases in which both EIT-observers detected spike potentials in the cyclic ROI information, CT scans of the chest revealed the presence of a PTX. The sudden drop of thoracic impedance may be explained by contact of the pericardium to the chest wall during systole in the case of the presence of a PTX located between the heart and the chest wall. This could be explained by either being directly related to the myocardium-chest wall contact, or by abnormal conduction of the current via the pulmonary catheter which was inserted in all of the animals in our study. Further studies are therefore required to examine the predictive value of spike potentials, to identify objective characteristics of spike potentials and to determine their exact pathophysiologic origin.

## Limitations of the study

There are several limitations of our study. Our chest trauma model was developed with the intention to induce severe lung injury resulting in traumatic ARDS. This extensive lung damage complicates interpretation of the EIT image due to the multitude of simultaneously arising pathologies leading to a ventilation defect in the EIT (hemothorax, atelectasis, pneumothorax). This may also partly explain the low correct detection rate of an existing PTX compared to otherwise healthy lungs with a PTX.

Second, the EIT device used was designed for use in humans, therefore image reconstruction is based on a thoracic model that is developed for human anatomy. If applied in animals with different thoracic anatomy, there is a mismatch between thoracic anatomy of the animal and the shapes used for image reconstruction, leading to a distorted image. This might also impair the visualization of a PTX. Individualized 3D-models for image reconstruction could further improve diagnostic quality of the EIT in the future.

EIT and CT image sets were randomly selected from 17 different animals, so within-subject effects may play a role on the results of the study.

Furthermore, the EIT-observers did not have access to additional clinical data that would be available in clinical routine. Additional information of the clinical state (chest X-rays, presence of chest tubes, patient history) may allow for a better interpretation of the EIT image. Moreover, all animals received bilateral chest drainage before a posttraumatic EIT was recorded in order to achieve hemodynamic stability. We were therefore not able to record an immediate posttraumatic EIT image showing an undrained large unilateral PTX or even a tension PTX.

## Conclusions

Taken together, our findings and the findings from Costa et al., Bhatia et al. and Cambiaghi et al. suggest that EIT is *theoretically* capable of providing the information needed for the diagnosis and monitoring of even small PTX. However, we found that, without reference information or dedicated image processing algorithms, the ad-hoc diagnosis or exclusion of a PTX using only the visual information currently provided on the display of the EIT devices cannot

exclude or detect a clinically relevant PTX with sufficient diagnostic safety or replace radiologic imaging.

## Supporting information

**S1 Table. Raw data depicting the results of the CT image analysis and the analysis of the corresponding EIT files.** EIT: Electrical Impedance Tomography; PTX: pneumothorax. (XLSX)

## Author Contributions

**Conceptualization:** Felix Girrbach, Tobias Landeck, Sören Hammermüller, Peter Salz, Hermann Wrigge, Andreas W. Reske.

**Formal analysis:** Felix Girrbach, Tobias Landeck, Dominic Schneider, Stefan U. Reske, Gunther Hempel, Udo Gottschaldt, Peter Salz, Andreas W. Reske.

**Investigation:** Felix Girrbach, Stefan U. Reske, Gunther Hempel, Sören Hammermüller, Udo Gottschaldt, Katharina Noreikat.

**Methodology:** Hermann Wrigge.

**Software:** Dominic Schneider, Peter Salz.

**Supervision:** Sebastian N. Stehr, Hermann Wrigge, Andreas W. Reske.

**Visualization:** Felix Girrbach.

**Writing – original draft:** Felix Girrbach, Tobias Landeck, Udo Gottschaldt, Andreas W. Reske.

**Writing – review & editing:** Felix Girrbach, Dominic Schneider, Gunther Hempel, Sören Hammermüller, Katharina Noreikat, Sebastian N. Stehr, Hermann Wrigge, Andreas W. Reske.

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
