## [Decision Letter · Decision Letter 0]

22 Aug 2019

PONE-D-19-19686

Detection of posttraumatic pneumothorax using Electrical Impedance Tomography - An observer-blinded study in pigs with blunt chest trauma

PLOS ONE

Dear Dr. Girrbach,

Thank you for submitting your manuscript to PLOS ONE. After careful consideration, we feel that it has merit but does not fully meet PLOS ONE’s publication criteria as it currently stands. Therefore, we invite you to submit a revised version of the manuscript that addresses the points raised during the review process.

We would appreciate receiving your revised manuscript by 30 days. To enhance the reproducibility of your results, we recommend that if applicable you deposit your laboratory protocols in protocols.io, where a protocol can be assigned its own identifier (DOI) such that it can be cited independently in the future. For instructions see: http://journals.plos.org/plosone/s/submission-guidelines#loc-laboratory-protocols

We look forward to receiving your revised manuscript.

Kind regards,

Federico Bilotta

Academic Editor

PLOS ONE

Journal Requirements:

2. To comply with PLOS ONE submissions requirements, please provide methods of sacrifice and the source of the animals in the Methods section of your manuscript.

Reviewers' comments:

Reviewer's Responses to Questions

**Comments to the Author**

1. Is the manuscript technically sound, and do the data support the conclusions?

Reviewer #1: Yes

Reviewer #2: Yes

2. Has the statistical analysis been performed appropriately and rigorously? 

Reviewer #1: Yes

Reviewer #2: Yes

3. Have the authors made all data underlying the findings in their manuscript fully available?

Reviewer #1: Yes

Reviewer #2: Yes

4. Is the manuscript presented in an intelligible fashion and written in standard English?

Reviewer #1: Yes

Reviewer #2: Yes

5. Review Comments to the Author

Reviewer #1: Thank you for the opportunity to revise this manuscript.I found this experimental study really interesting, with a robust methodology and an important and clear message, possibly easy to apply in the clinical practice. Also, the authors should be congratulated because of the attempt to find tools able to provide clinical findings at the patients bedside.The manuscript is well written.

I have some minor suggestions:

There are some typo mistakes, please revise the entire manuscript for English and grammar

Introduction: Line 69-70. This information (…may convert to tension pneumothorax..) is redundant

Please add in the introduction some informations about the principles of EIT

The discussion section should be structured according to the order of the results. In this form it is really confusing.

Reviewer #2: 1. ABSTRACT.

- Abstract appropriately summarize the manuscript.

- There aren´t discrepancies between the Abstract and the remainder of the manuscript.

- The Abstract can be understood without reading the manuscript.

2. BACKGROUND/AIM

- The Introduction is concise.

- The purpose of the study is well defined.

- The authors provide a rationale for performing the study based on a review of the medical literature with an appropriate length.

- This manuscript is Original Research, with a well-defined hypothesis.

3. MATERIALS AND METHODS

- Please, define and explain more accurate the study design.

4. RESULTS

- The results are clearly explained.

- The results are reasonable and expected.

- There aren´t results introduced that are not preceded by an appropriate discussion in the Methods section.

5. DISCUSSION

- The discussion is concise.

- Their research question was answered.

- Authors’ conclusions are justified by the results found in the study.

6. FIGURES

- Figure is appropriate and it is appropriately labeled.

- Adequately show the important results.

7. TABLES (Supl)

- Appropriately describe the results.

8. REFERENCES

- The reference list follows the format for the journal.

6. PLOS authors have the option to publish the peer review history of their article (what does this mean?). If published, this will include your full peer review and any attached files.

Reviewer #1: No

Reviewer #2: Yes: Rafael Badenes

---

## [Author Response · Author response to Decision Letter 0]

31 Oct 2019

RESPONSE TO REVIEWER #1

Thank you for your valuable suggestions. We changed the manuscript accordingly:

- The entire manuscript was revised concerning English and grammar

- The redundant information in line 69-70 was deleted. According, we changed the sentence in line 60 as follows: “Undiagnosed, occult pneumothoraces may convert to tension pneumothorax - particularly after initiating mechanical ventilation - and require emergency thoracostomy (2, 3).”

- As suggested, we inserted some information about the principles of EIT in the introduction

- The discussion section was restructured as suggested

RESPONSE TO REVIEWER #2

Thank you also for your helpful suggestions. We changed the manuscript according to your suggestion:

- The study design is defined and explained more accurate in the revised manuscript. The first paragraph of the methods section was changed as follows:

“In compliance with the 3R’s for reduction of animals in research (https://www.nc3rs.org.uk/the-3rs), we used suitable data from an existing institutional database in this experimental, observer-blinded diagnostic study. The original experiment was approved by the governmental animal ethics committee (Landesdirection Leipzig, reference number TVV38/11). Pigs (German Landrace) were received from a conventional pig-breeding farm and were kept at the Large Animal Clinic for Surgery, Faculty of Veterinary Medicine, University of Leipzig in a species-specific environment. 

From this database, we randomly obtained 100 EIT files and the corresponding CT data from 17 different pigs. The files were anonymized and observers were blinded to all other information. Two observers (specialists in anesthesiology and intensive care) analyzed the randomly obtained 100 EIT files, while a third observer (specialist in radiology) analyzed the corresponding CT images. “

---

## [Decision Letter · Decision Letter 1]

20 Dec 2019

Detection of posttraumatic pneumothorax using Electrical Impedance Tomography - An observer-blinded study in pigs with blunt chest trauma

PONE-D-19-19686R1

Dear Dr. Girrbach,

We are pleased to inform you that your manuscript has been judged scientifically suitable for publication and will be formally accepted for publication once it complies with all outstanding technical requirements.

With kind regards,

Federico Bilotta

Academic Editor

PLOS ONE

Additional Editor Comments (optional):

PONE-D-19-19686R1

In this observed study, the Authors analyzed an existing database of 17 mechanically ventilated pigs, which had sustained standardized blunt chest trauma and had undergone repeated thoracic CT and EIT.

100 corresponding EIT/CT datasets were randomly chosen from the database and anonymized. Two independent and blinded observers analyzed the EIT data for presence and location of PTX. Analysis of the corresponding CTs by a radiologist served as reference.

Results 87/100 cases had at least one PTX detected by CT. Fourty-two cases showed a PTX > 20% of the sternovertebral diameter (PTXtrans20), whereas 52/100 PTX showed a PTX>3 cm in the craniocaudal diameter (PTXcc3), with 20 cases showing both a PTXtranscc and a PTXcc3. We found a very low agreement between both EIT observers considering the classification overall PTX/noPTX (κ=0.09, p=0.183).

The Authors concluded that, multimodal imaging approaches, technical improvements and image postprocessing algorithms might improve the performance of EIT for diagnosing PTX in the future.

Comments

Reviewer 1: The authors have addressed my concerns and should be congratulated for the effort in the revisions. Thank you. Accept

Reviewer 2: The authors have satisfactorily responded to all my questions and made the necessary changes to the manuscript. Accept

Academic editor: considering changing and positive feedback from the reviewers I’m informed you that your manuscript is suitable for publication.

Reviewers' comments:

Reviewer's Responses to Questions

**Comments to the Author**

1. If the authors have adequately addressed your comments raised in a previous round of review and you feel that this manuscript is now acceptable for publication, you may indicate that here to bypass the “Comments to the Author” section, enter your conflict of interest statement in the “Confidential to Editor” section, and submit your "Accept" recommendation.

Reviewer #1: All comments have been addressed

Reviewer #2: All comments have been addressed

2. Is the manuscript technically sound, and do the data support the conclusions?

Reviewer #1: Yes

Reviewer #2: Yes

3. Has the statistical analysis been performed appropriately and rigorously? 

Reviewer #1: Yes

Reviewer #2: Yes

4. Have the authors made all data underlying the findings in their manuscript fully available?

Reviewer #1: No

Reviewer #2: Yes

5. Is the manuscript presented in an intelligible fashion and written in standard English?

Reviewer #1: Yes

Reviewer #2: Yes

6. Review Comments to the Author

Reviewer #1: The authors have addressed my concerns and should be congratulated for the effort in the revisions. Thank you.

Reviewer #2: The authors have satisfactorily responded to all my questions and made the necessary changes to the manuscript.

7. PLOS authors have the option to publish the peer review history of their article (what does this mean?). If published, this will include your full peer review and any attached files.

Reviewer #1: No

Reviewer #2: Yes: Rafael Badenes

---

## [Editor Report · Acceptance letter]

26 Dec 2019

PONE-D-19-19686R1 

Detection of posttraumatic pneumothorax using Electrical Impedance Tomography - An observer-blinded study in pigs with blunt chest trauma 

Dear Dr. Girrbach:

I am pleased to inform you that your manuscript has been deemed suitable for publication in PLOS ONE. Congratulations! Your manuscript is now with our production department. 

With kind regards,

on behalf of

Dr. Federico Bilotta 

Academic Editor

PLOS ONE